# Synergistic Flavor Modulation and Functional Enhancement of Douchiba via Compounding with *Bacillus subtilis*-Fermented Adlay

**DOI:** 10.3390/foods14172976

**Published:** 2025-08-26

**Authors:** Lian Peng, Yongjun Wu, Anyan Wen, Haiying Zeng, Likang Qin

**Affiliations:** 1School of Liquor and Food Engineering, Guizhou University, Guiyang 550025, China; lianpeng2002@163.com (L.P.); hyzeng1@gzu.edu.cn (H.Z.); lkqin@gzu.edu.cn (L.Q.); 2Key Laboratory of Plant Resource Conservation and Germplasm Innovation in Mountainous Region (Ministry of Education), College of Life Sciences/Institute of Agro-Bioengineering, Guizhou University, Guiyang 550025, China; wyjbio@163.com; 3Key Laboratory of Agricultural and Animal Products Storage and Processing of Guizhou Province, Guiyang 550025, China; 4National & Local Joint Engineering Research Center for the Exploitation of Homology Resources of Medicine and Food, Guiyang 550025, China

**Keywords:** Entropy Method, E-tongue, compound seasoning, active ingredient, flavor substances

## Abstract

Traditional Douchiba (DCB), a bacterial-type fermented soybean condiment, suffers from pronounced bitterness and limited functional attributes, hindering its broader application. To address these challenges, this study innovatively compounded matured *Bacillus subtilis*-fermented adlay (BFA) with DCB at varying ratios to develop a fermented adlay-DCB seasoning (FADS). Key physicochemical, nutritional, functional, and sensory parameters were systematically analyzed, and a multidimensional quality evaluation system was established via the Entropy Method for composite scoring. Results revealed that BFA integration enhanced the brightness and increased the content of total triterpenoid (by 16-fold) and γ-aminobutyric acid (by 9-fold) in FADS. Notably, electronic tongue analysis demonstrated that BFA significantly reduced the bitterness, after-bitterness, and saltiness intensities of DCB, achieving maximum reductions of 90.12% for bitterness and 87.63% for after-bitterness. Meanwhile, GC-MS profiling identified 89 volatile compounds, with pyrazines, alcohols, and acids as the primary volatile components in FADS. Additionally, the S4 sample (the BFA:DCB ratio = 6:4) achieved the highest composite score (0.64), with pyrazines contributing 0.13 points to the evaluation. In summary, BFA not only significantly mitigated bitterness in DCB but also substantially enhanced its bioactive properties. The results offer a scientific basis for the flavor improvement of fermented seasonings.

## 1. Introduction

Douchi, a traditional fermented food product derived from soybeans in China, can be used not only as a main course but also as a seasoning [1]. Based on the predominant fermentative microorganisms, Douchi is categorized into four types, namely *Aspergillus*-type, *Mucor*-type, *Rhizopus*-type, and bacterial-type [2]. In bacterial-type Douchi, the microbiota is dominated by *Bacillus* spp., *Micrococcus* spp., and lactic acid bacteria [3]. The production process of bacterial-type Douchi undergoes an initial aerobic fermentation phase where bacterial proteases mediate the degradation of soybean proteins. Subsequently, the growth rate of microorganisms is slowed down by altering the microbial growth environment using salt and liquor. Additionally, flavor substances are increased, and the beany odor is removed by adding ginger powder and pepper powder during anaerobic post-fermentation. Finally, the product is dried to obtain the finished Douchi [4]. The production of Douchi embodies the artisanal heritage that has evolved over centuries, thereby endowing it with distinctive organoleptic properties and unique flavor characteristics.

Douchiba (DCB), a derivative of bacterial-type Douchi, is a unique traditional fermented condiment indigenous to Guizhou Province, China. It has the ability to regulate appetite, alleviate fatigue, improve insomnia and offer antioxidant qualities [5]. The traditional production of DCB involves a succession of steps including soaking, steaming, koji-making, salting, fermentation, grinding, sun-drying, molding, and after-ripening, spanning 12−18 months [6]. This complicated processing endows distinctive product characteristics, a rich and long-lasting flavor, and a dark and oily appearance [7]. However, the free amino acids in Douchiba were predominantly bitter amino acids (accounting for approximately 40% of total free amino acids), and the contents of hydrophobic bitter peptides (molecular weights of 150–200 Da and <150 Da) were increased with prolonged fermentation time [8]. Furthermore, electronic tongue analysis confirmed that pronounced bitterness and after-bitterness were prominent taste characteristics [9]. Consequently, mitigating bitterness, while enhancing overall flavor complexity, represents a critical challenge that needs to be addressed.

One promising strategy to modulate undesirable flavors in fermented foods involves compounding with complementary flavor-rich ingredients. Previous studies have explored the co-fermentation of adlay with soybeans by *Bacillus subtilis* to improve flavor profiles and functional properties [10,11]. Although these studies demonstrated the potential of adlay and soybean co-fermentation, the distinct approach that the direct compounding of matured *Bacillus subtilis*-fermented adlay (BFA) with DCB remains underexplored. BFA has been confirmed to increase levels of valuable compounds, including tetramethylpyrazine (TMP), γ-aminobutyric acid (GABA), free amino acids, triterpenes, and phenolics. Specifically, the TMP content in BFA was 200 times higher than that in soybeans fermented with the same strain, and the GABA content was 12 times higher than that in unfermented adlay [12]. Meanwhile, extracts from BFA exhibited potent anti-proliferative activity against human leukemia K562 cells and human non-small cell lung cancer A549 cells [13]. Thus, BFA emerges as a promising substrate for developing high-value functional seasonings.

Therefore, this study innovatively compounded BFA as a flavor-regulating substrate with DCB, at varying ratios (the BFA:DCB ratio = 3:7 to 7:3), to develop a novel fermented adlay-DCB seasoning (FADS). The primary objectives were to evaluate the impact of BFA incorporation on the nutritional and functional components of FADS and analyze the changes in taste profiles and volatile flavor compound composition resulting from the BFA/DCB interaction. Finally, a quality evaluation system was established using the Entropy Method to determine the optimal BFA-to-DCB ratio that achieved the best balance between physicochemical properties and sensory acceptability. The study not only resolved the contradiction between functional enhancement and flavor degradation but also provided a scientific basis for the development of a new generation of fermented condiments.

## 2. Materials and Methods

### 2.1. Materials and Reagents

*B. subtilis* BJ3-2 was a kind gift from Dr. Yongjun Wu (College of Life Sciences, Guizhou University, Guizhou, China). Douchiba was sampled from Huang Fuyuan Food Factory, Dafang County, Bijie City, Guizhou Province, China. Adlay was obtained from Guizhou Renxin Agricultural Development Co., Ltd. (Guizhou, China). Ethanol, sodium hydroxide, hydrochloric acid, gallic acid, rutin, oleanolic acid, and γ-aminobutyric acid were purchased from Sigma Chemical Co. (St. Louis, MO, USA). All other chemicals and reagents were of analytical grade and purchased from Sinopharm Chemical Reagent Co., Ltd. (Suzhou, China) and were of analytical grade.

### 2.2. Preparation of FADS

Fermented adlay was prepared according to the previous method described by Wen et al. [12]. BFA and DCB were freeze-dried (SCIENTZ-18N, Shanghai, China). They were then compounded according to the following ratios (g/g): 0:1 (CK), 3:7 (S1), 4:6 (S2), 5:5 (S3), 6:4 (S4), and 7:3 (S5), and ground into a powder (60 mesh). The powder obtained from the different combinations was stored in vacuum bags. In addition, the obtained compound seasonings were stored for later use at 4 °C in a refrigerator to preserve volatile and functional compounds prior to analysis.

### 2.3. Determination of Color Difference

L*, a*, and b* values were determined by colorimeter (CR-410, Japan) following the method described by Subtain et al. [14]. The calculation of the total color difference (ΔE) was performed as follows:$$∆E=\sqrt{\left(a^{2})+(b^{2})+(L^{2}\right)}$$

### 2.4. Determination of Water Holding Capacity (WHC) and Oil Holding Capacity (OHC)

Water holding capacity (WHC) was determined as follows. Firstly, 1.5 g of the sample was weighed in a centrifuge tube, and then, water was added gradually. The mixture was stirred with a glass rod until the slurry was formed, but water did not precipitate. The slurry was then centrifuged for 10 min at 25 °C and 4000 r/min (H2-16KR, Changsha, Hunan, China) until little water precipitated. Finally, the obtained mixture was weighed. The WHC was calculated as follows:(1)$$WHC(\%)=M_{1}-M_{2}/M$$
where M_1_ is the total mass of the sample and the centrifuge tube; M_2_ is the total mass of the sample and the centrifuge tube obtained after centrifugation; M is the sample mass.

Oil holding capacity (OHC) was determined as follows. Firstly, 5.0 g of the sample was weighed, followed by the addition of 30 mL of peanut oil. After heating in boiling water for 20 min, the mixture was centrifuged for 15 min at 3000 rpm. After the top layer of oil was poured out, the centrifuge tube was inversed for another 15 min and then weighed. The OHC was calculated as follows:(2)$$OHC(\%)=S_{1}-S_{2}/S$$
where S_1_ is the total mass of the sample and the centrifuge tube; S_2_ is the total mass of the sample and the centrifuge tube obtained after centrifugation; S is the sample mass.

### 2.5. Determination of Nutritional Components

The starch, fat, protein, ash, and moisture contents were determined with the method of Fu et al. [15].

### 2.6. Determination of Functional Components

Total flavonoids and total phenols were determined with the method proposed by Almaghlouth et al. [16]; triterpenes were determined with the vanillin-perchloric acid colorimetric reaction method [17]; GABA was determined with the method proposed by Wen et al. [12].

### 2.7. Determination of the Free Amino Acid

Free amino acids were determined according to the method of Qiao et al. [18].

### 2.8. Determination of Taste Substances

Taste substances were determined using the SA 402 B electronic tongue (Insent, Kanagawa, Japan) according to the method described by Liu et al. [19] with minor modifications. Firstly, 5.0 g of the sample was weighed, diluted by 15 times with deionized water, ground in a grinder, ultrasonically sonicated for 90 min and centrifuged (8000× *g*, 5 min). After the supernatant was removed, the obtained mixture was used for electronic tongue analysis.

### 2.9. Determination of Volatile Flavor Substances

Volatile compounds were determined according to the previous study with some slight modifications [8].

Volatile compounds in FADS were analyzed based on a HS-SPME-GC-MS (Pegasus BT, LECO, USA). With an aged 50/30 μm CAR/PDMS/DVB extraction head, the samples were separated into 20 mL headspace vials, adsorbed for 30 min at 60 °C and then desorbed for 3 min at 250 °C. Finally, the data were acquired.

GC conditions were set as follows: DB-Wax Chromatography Column (30 m × 0.25 mm, 0.25 μm). Heating program: initial temperature (40 °C) for 3 min, final temperature in the oven (230 °C) for 5 min, and heating rate (10 °C/min).

MS conditions were set as follows: ionization mode (El+), electron energy (70 eV), surface temperature (200 °C), interface temperature (250 °C), detector voltage (2000 V), and emission current (1 mA).

### 2.10. Data Analysis

In each experiment, three parallel samples were set. The experimental data were expressed as mean ± SD. The experimental data were analyzed and processed in SPSS 27.0 software (SPSS Inc., Chicago, IL, USA) and displayed in Origin 2024 software (OriginLab Corporation, Northampton, MA, USA). The differences among the results were analyzed by Duncan’s test and analysis of variance (ANOVA) (*p <* 0.05).

## 3. Results and Discussion

### 3.1. Color Difference Analysis of FADS

Color affects consumers’ acceptance of products and evaluation of product quality [20]. After freeze-drying and grinding, DCB exhibited a dark brown color. As the content of BFA increased, the color of FADS became progressively lighter (Figure 1A). This observation was quantitatively supported by colorimeter measurements. As shown in Figure 1B, the L-value demonstrated a positive correlation with BFA addition. As BFA proportions increased, the a-value showed a gradual decrease while the b-value exhibited a sustained increase. Notably, BFA displayed a bright white appearance, contrasting sharply with DCB’s dark brown coloration. Consequently, mixed samples showed distinct color profiles, namely, S1 retained higher a-values, whereas S5 exhibited elevated b-values. Additionally, the total color difference (ΔE) of the FADS ranged from 32.63 to 37.16, which varied from DCB. Especially in mixed samples with a high content of BFA (S3, S4, and S5), the difference was more pronounced. Overall, this change improved the sensory acceptability of the compound seasoning to some extent.

### 3.2. Basic Quality Analysis in FADS

WHC and OHC of FADS are the core indicators of the functional properties and directly determine the texture and performance of the final product [21]. The formation of protein−starch complexes and the structure of polysaccharide networks are strongly linked to WHC, which indicates the capacity to bind water [22]. OHC describes the binding efficiency between lipids and protein matrix and has a direct impact on the flavor release and palatability of the final product [23]. With the increase in the addition of BFA, WHC of FADS firstly declined, then increased and then declined again, and it peaked in S4 (90.85%) and S1 (93.8%) (Figure 2A). The higher WHC of S1 could be interpreted as follows. The protease in DCB promoted the hydrolysis of surface proteins of adlay starch granules so that more hydrophilic groups were exposed [24]. On the contrary, S2 and S3 had the lower WHC, probably because a proper amount of adlay fermentation metabolites (e.g., organic acids) weakened the amylose−protein network through hydrogen-bonding competition. With the increase in the fraction of BFA, OHC of FADS gradually increased and peaked in S5 (0.23%). This change was related to the altered protein conformation caused by the content of DCB. A small amount of DCB could improve lipid binding by increasing hydrophobicity. However, when the proportion of DCB was too high, it caused excessive protein aggregation, resulting in the formation of dense structures that hindered lipid penetration [25].

The starch content in FADS samples exhibited an increasing tendency and peaked at 66.41% in S5 (Figure 2B). The protein content showed a decreasing trend and peaked at 25.60% in S1. The starch and protein contents of BFA were 70% and 15%, respectively, whereas the starch and protein contents of DCB were 40% and 25%, respectively. Consequently, with the increase in the fraction of BFA, the starch content of FADS increased, while the protein content decreased. Both DCB and BFA had low fat content, so there was no significant change in FADS samples. The ash content between samples ranged from 3.06% to 3.46%, among which the CK group had the highest content. Moisture content affects the storage stability of products. The moisture content of the six samples ranged from 2.89% to 3.26%, with S3 exhibiting the lowest moisture content.

In summary, these synergistic effects between the components in DCB and BFA not only optimized the nutrient ratios but also improved WHC and OHC of the composite system through protein−starch or protein−lipid interactions.

### 3.3. Contents of Active Ingredients

Flavonoids and polyphenols are one type of the primary antioxidant active ingredients in DCB and also have anti-inflammatory, anticancer, hypoglycemic, and immunity-boosting properties [26,27]. The total flavonoid content and total phenol content of FADS firstly decreased and then increased as the proportion of BFA increased (Figure 3A). The highest contents of flavonoids and phenols were found in S4 (16.01 mg RE/100 g) and CK (2.06 mg GAE/100 g), respectively. The addition of BFA decreased the total flavonoid content and total phenol content of FADS because the flavonoid and polyphenol contents in fermented adlay were lower than those in DCB.

Triterpenoids have anticancer, antiviral, hypoglycemic, and hypolipidemic effects [28]. GABA, a four-carbon non-protein amino acid, is crucial for controlling the central nervous system, reducing blood pressure and cholesterol, and relaxing and relieving excitement [29]. As the proportion of BFA increased, the contents of total triterpene and GABA in FADS gradually increased (Figure 3B). In particular, S5 had the highest levels of total triterpene (10.39 mg/g) and GABA (14.77 mg/g), which were 16 times and 9 times higher than that in DCB, respectively. The contents of total triterpene and GABA in raw adlay were 8.0 mg/g and 2.5 mg/g, respectively, and increased to 17.38 mg/g and 30.14 mg/g, respectively, after fermentation [12]. Zhang et al. [30] found the content of GABA in raw soybeans was not detected, and after fermentation, it increased to 0.43 mg/g. A study had shown that the content of DDMP (2,3-dihydro-2,5-dihydroxy-6-methyl-4H-pyran-4-one, one type of soyasaponins) in raw soybeans was 481.3 μmol/100 g DB, which increased to 532.1 μmol/100 g DB after fermentation [31]. Although triterpenoids and GABA, which are low in soybeans, can also be increased by fermentation, their level is still less than that of fermented adlay. As a result, the proportion of BFA directly determined the contents of total triterpene and GABA in FADS. GABA could reduce pain and anxiety; therefore, it could treat neurological disorders such as depression. In addition, it also activates GABA_A_ and GABA_B_ receptors in pancreatic β cells. Thereby, it could promote insulin release and treat diabetes [32]. Triterpenoids, particularly lupinol and betulin, exhibit strong anti-inflammatory and antibacterial properties [33]. Therefore, FADS, which is rich in triterpenoids, could compensate for its low flavonoids and polyphenols content. Overall, adding BFA enhanced the physiological activities of the seasoning, such as blood pressure reduction and anti-anxiety effects, and strengthened its anti-inflammatory and anticancer properties.

### 3.4. Contents of Free Amino Acids

Free amino acids include both essential amino acids and non-essential amino acids. They are crucial for product taste, acting as precursors to volatile flavor compounds and being essential for food palatability [34]. As shown in Table 1, a total of 16 amino acids were detected in all samples, including 7 essential amino acids. The composite seasonings S3 and S4 had the highest level of total free amino acids (2.86 g/100 g). Leucine, as the primary free amino acid in FADS, has the efficacy of facilitating muscle repair, regulating blood sugar, and supplying energy to tissues [35]. The increased release of leucine from adlay or soybean resulted from the targeted hydrolysis of proteins by alkaline proteases secreted by *B. subtilis* during fermentation [36]. Leucine was the most abundant free amino acid in adlay fermented with *B. subtilis,* according to the report by Wen et al. [37]. The amino acids in grains and legumes are usually complementary [38]. Thus, the content of essential amino acids in FADS was more balanced as a result of amino acid complementation between BFA (methionine and leucine) and DCB (lysine).

By classifying the detected free amino acids into four groups based on their flavor-presenting characteristics, it was found that bitter (1.31−1.78 g/100 g) and sweet amino acids (0.53−0.65 g/100 g) were the major flavor-presenting amino acids in FADS. Although the content of bitter amino acids is relatively high, the presence of bitter amino acids plays an important role in prolonging taste and enhancing complexity and richness [8]. The content of umami-presenting amino acids was 0.36−0.46 g/100 g. Especially, the content of glutamic acid was high, so FADS had an intense umami. Meanwhile, the umami of FADS was further enhanced by the synergistic effect of sweet amino acids on umami perception [39].

In conclusion, the free amino acid profile of FADS achieved the dual objectives of enhancing flavor quality and optimizing nutritional functions.

### 3.5. Analysis of Taste Substances

Figure 4A displays the taste intensity and taste characteristics of FADS. Umami, sourness, salty flavor, bitterness, and after-bitterness among the six samples showed significant differences, but the difference in richness or sweetness was not significant. DCB (CK) exhibited distinctive taste characteristics, including high levels of umami, saltiness, bitterness, and after-bitterness. Although DCB had a distinctive umami flavor, it also had a strong bitter taste. Therefore, the potential application of DCB as a seasoning was certainly limited. Fermented adlay exhibited high acidity [40]. Research showed that in acidic environments, glutamic acid could combine with sodium ions to form monosodium glutamate, which enhanced the umami characteristic [41]. Thus, the addition of BFA could enhance the release of umami flavor from DCB.

Hydrophobic peptides, bitter amino acids, isoflavonoids, and saponins are common compounds with bitter properties. The bitterness of FADS was directly correlated with DCB proportion because these substances were produced and increased in the fermentation of DCB [10]. However, the addition of BFA could improve the bitterness and after-bitterness of DCB. As shown in Figure 4B,C, the values of bitterness and after-bitterness in the composite seasonings were lower than those in DCB. Specifically, when 60% BFA was incorporated (S4), the values of bitterness and after-bitterness decreased by 90.12% and 87.71%, respectively, compared to DCB. Meanwhile, at 70% BFA supplementation (S5), the corresponding reductions reached 80.94% for bitterness and 87.63% for after-bitterness. This occurred because the bitter peptides in DCB may be complexed by the organic acids (lactic, malic, citric, etc.) in BFA, which would lessen the bitterness of the compounded system. The mechanism was as follows: the hydroxyl groups (-OH) in organic acids could establish hydrogen bonds with the hydrophobic amino acid residues in bitter peptides, while hydrophobic amino acid residues contributed most significantly to the bitterness of bitter peptides [42]. This hydrogen bonding interaction altered the secondary structure of bitter peptides, which partially shielded the hydrophobic amino acid residues and reduced their contact with bitter receptors, thereby reducing bitterness.

Free amino acid content and electronic tongue are both important indicators of taste. Correlation analysis was performed to further investigate the correlation between flavor amino acids and electronic tongue taste characteristics in different mixed samples. Figure 4D illustrates how the flavor amino acids correlate with the taste qualities of the electronic tongue. Bitterness and after-bitterness were correlated positively with serine, glycine, isoleucine, and lysine. Among them, bitterness and isoleucine showed a significant correlation, while aftertaste showed a significant correlation with isoleucine and glycine (*p <* 0.01). Therefore, reducing the content of isoleucine and glycine could improve the bitter taste characteristics of DCB. Compared with other samples, CK had the greatest levels of glycine and isoleucine. The amount of isoleucine and glycine in FADS steadily dropped with the addition of BFA, and sample S5 had the lowest levels of these two amino acids (Table 1). This suggested that the bitterness of DCB could be regulated by adding BFA.

In conclusion, the incorporation of BFA effectively improved the taste characteristics of DCB, with notable reductions observed in bitterness, after-bitterness, and saltiness intensity.

### 3.6. Relative Content of Volatile Flavor Substances

A total of 89 volatile flavor components were detected in FADS, including 18 pyrazines, 11 alcohols, 11 esters, 8 acids, 11 aldehydes, 13 ketones, 5 amines, 4 phenols, and 8 others (Figure 5A–D). Among the five composite seasoning sample groups, pyrazines exhibited the highest relative content (28.60–42.68%), followed by alcohols (4.11–31.84%) and acids (4.04–18.98%). These findings indicate that pyrazines, alcohols, and acids constituted the primary flavor-contributing compounds in FADS.

Pyrazines with a very low sensory threshold and nutty aroma characteristics, as the key contributors to the characteristic flavor of DCB, dominated the formation of its flavor profile [43]. DCB contained various kinds of pyrazines (17 kinds). However, BFA had the higher relative contents of tetramethylpyrazine (TMP), 2,3,5-trimethyl-6-ethylpyrazine and trimethylpyrazine (Figure 5C).

Among these compounds, TMP, a core functional component, had the functions of inhibiting platelet aggregation, regulating lipid metabolism, inducing apoptosis of tumor cells, and other physiological functions [44]. It had two biosynthesis pathways. Firstly, via the metabolic pathway, *B. subtilis* decomposed glucose into pyruvate, which was then further transformed into acetoin. Meanwhile, microorganisms released proteases to decompose adlay proteins to produce free amino acids. These amino acids were then deaminated to produce ammonium, which condensed with acetoin to produce TMP under certain conditions [45]. Through Amadori rearrangement, reducing sugars and α-amino acids were transformed into α-dicarbonyl compounds during the drying process of DCB. Through Strecker degradation, these compounds were converted into α-amino ketones, which were then polymerized into TMP [46]. Meanwhile, TMP also endowed the product with the cocoa and roasting aroma. Furthermore, the unique flavors of fermented adlay were combined with the cocoa and nutty scents from 2,5-dimethylpyrazine and trimethylpyrazine so as to greatly enhance the richness and harmony of the flavor.

Alcohols largely determine the flavor of FADS. Alcohol production was mostly dependent on hydroperoxide degradation from fatty acid β-oxidation and carbonyl compound reduction [47].

The highest relative total contents of alcohols (31.84%) and 2,3-butanediol (26.07%) were found in Sample S5 (Figure 5C). The biosynthesis pathway of 2,3-butanediol was well defined. *B. subtilis* converted pyruvate into acetoin with acetolactic acid decarboxylase and then reduced with NADH-dependent 2,3-butanediol dehydrogenase to produce 2,3-butanediol [48]. The fruity and creamy characteristics of 2,3-butanediol enhanced the flavor richness of FADS. Phenylethanol was detected in all samples. It contributed a rosy-like and citrus aroma to FADS, while counteracting the intense odors of other strong flavorings (e.g., bitterness), thereby creating a softer flavor profile. During fermentation, *B. subtilis* converted L-phenylalanine via the Ehrlich pathway, involving transamination, decarboxylation, and reduction, to produce phenylethanol [49].

Acids, as one kind of the main volatile components of the taste system of FADS, have a considerable effect on sensory attributes due to their low-threshold volatility features. As shown in Figure 5A, the relative content of acids in Sample S5 was high (18.98%).

The main carboxylic acids were 3-methylbutyric acid (4.68~6.52%), acetic acid (2.40~5.63%), and butyric acid (0.65~2.21%). 3-Methylbutyric acid, which endowed the product with fruity and creamy scents and other qualities, was produced by the lipolysis of medium-chain fatty acids with six or more carbons or the degradation of leucine and valine [50]. Short-chain fatty acids, such as acetic acid, butyric acid, and caproic acid, can impart the characteristic sourness to seasonings. As the proportion of BFA increased, the contents of short-chain fatty acids gradually increased in FADS (Figure 5C). These acids could not only improve the bitterness of DCB but also interact with pyrazines and alcohols to balance the overall flavor of the composite seasonings.

Esters were the primary scent components of the flavor system of FADS, and their high volatility and low threshold significantly affected the fruity and floral olfactory qualities of products. After the addition of BFA, the ester kinds and contents all increased compared with DCB (Figure 5A). Among those samples, the types in S4 samples increased to 10, and the relative content in S2 samples reached the highest (9.97%).

In FADS samples, ethyl esters were relatively abundant in volatile compounds, especially ethyl acetate and vinyl acetate. Fermented adlay contained high levels of acetic acid, which underwent esterification reactions with alcohols to generate ethyl ester compounds, thereby enhancing the overall hedonic quality of FADS aroma. Notably, propyl 2-methylpropionate was not detected in DCB. However, probably due to the synergistic effect of fermented adlay and DCB, its content was high in the samples S3, S4, and S5. Propyl 2-methylpropionate had the low threshold and pineapple-like fruity flavor and sweet scent, which significantly enhanced the aroma complexity and attractiveness of FADS. In addition, esters also stabilized the volatility of pyrazines (e.g., TMP) through hydrophobic interactions, thus resulting in longer-lasting flavors.

### 3.7. Comprehensive Quality Evaluation of FADS

The data of the indices were standardized, and the compound seasonings were rated (Figure 6). The scores were based on WHC, OHC, flavonoids, polyphenols, triterpenes, GABA, umami amino acids, sweet amino acids, bitter amino acids, odorless amino acids, pyrazines, alcohols, acids, and esters. The results showed that S4 samples had the highest overall score (0.64), indicating that the addition of fermented adlay effectively improved the deficiencies of DCB. Specifically, the score of pyrazine compounds in S4 was 0.13, which contributed a cocoa flavor and nutty scent to DCB seasoning. This content level effectively harmonized the overall flavor profile and enhanced sensory acceptance. The analyzed results of the Entropy Method indicated that S4 samples had the highest overall score.

## 4. Conclusions

This study developed a functional fermented seasoning (FADS) through strategic compounding of BFA and DCB, successfully improving both sensory attributes and functional properties of sensory quality and bioactive components. The addition of BFA significantly increased the total triterpenoid and GABA contents in FADS, with sample S5 showing triterpenoid and GABA levels of 10.39 mg/g and 14.77 mg/g, respectively. Meanwhile, the addition of BFA reduced the intensity values of bitterness, bitter aftertaste, and saltiness in DCB, with maximum reductions of 90.12% and 87.63% observed for bitterness and after-bitterness, respectively. Furthermore, BFA increased the varieties and contents of key flavor compounds in FADS, including pyrazines, alcohols, acids, and esters, which synergistically improved the complexity and harmony of the flavor profile. The Entropy Method analysis revealed that sample S4 had the highest overall score, demonstrating the optimal balance between flavor and functional attributes. In conclusion, our research showed that FADS improved both functional and sensory qualities in concert. This improvement stemmed from complementary amino acids, bidirectional taste compensation, and synergistic effects of flavor components between BFA and DCB. These findings provide a fundamental theoretical basis for optimizing flavor profiles in fermented seasonings.

Future research should investigate the molecular interactions between organic acids in BFA and bitter peptides in DCB to clarify the bitterness-suppressing mechanism of BFA in fermented seasonings. In addition, standardized production protocols need to be developed, and the long-term storage stability and processing adaptability of FADS require systematic investigation.

## Figures and Tables

**Figure 1 foods-14-02976-f001:**
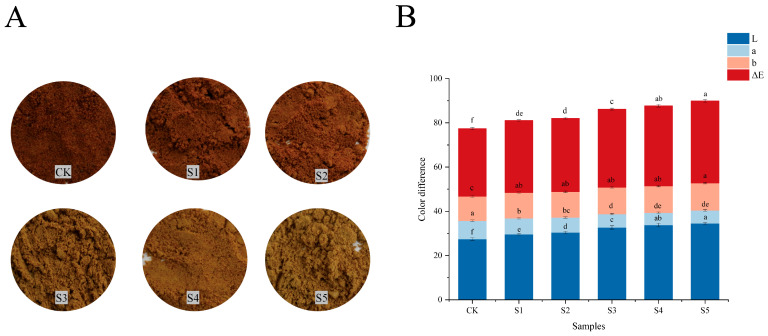
(**A**) Graphs of samples with different ratios. (**B**) Color difference results of FADS. CK—DCB; S1—the BFA:DCB ratio was 3:7; S2—the BFA:DCB ratio was 4:6; S3—the BFA:DCB ratio was 5:5; S4—the BFA:DCB ratio was 6:4; S5—the BFA:DCB ratio was 7:3. The different letters indicate significant differences (*p* < 0.05).

**Figure 2 foods-14-02976-f002:**
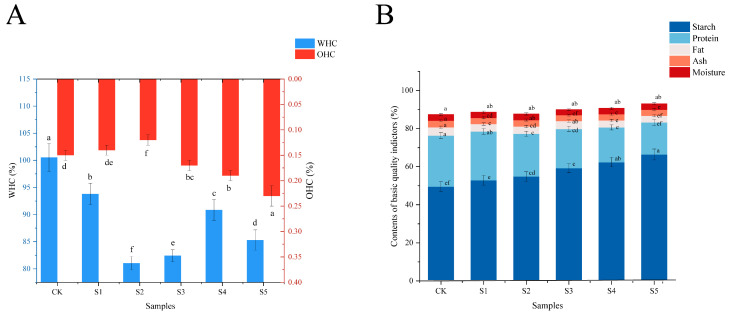
(**A**) WHC and OHC in FADS; (**B**) fat, starch, protein ash, and moisture contents in FADS. CK—DCB; S1—the BFA:DCB ratio was 3:7; S2—the BFA:DCB ratio was 4:6; S3—the BFA:DCB ratio was 5:5; S4—the BFA:DCB ratio was 6:4; S5—the BFA:DCB ratio was 7:3. The different letters indicate significant differences (*p* < 0.05).

**Figure 3 foods-14-02976-f003:**
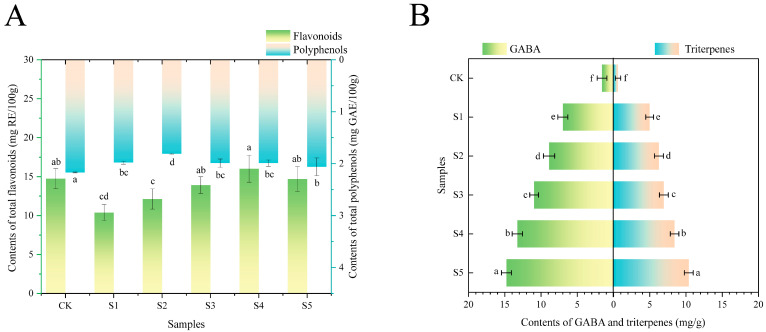
Total flavonoids, total polyphenols (**A**), triterpenes, and GABA (**B**) contents in FADS. CK—DCB; S1—the BFA:DCB ratio was 3:7; S2—the BFA:DCB ratio was 4:6; S3—the BFA:DCB ratio was 5:5; S4—the BFA:DCB ratio was 6:4; S5—the BFA:DCB ratio was 7:3. The different letters indicate significant differences (*p* < 0.05).

**Figure 4 foods-14-02976-f004:**
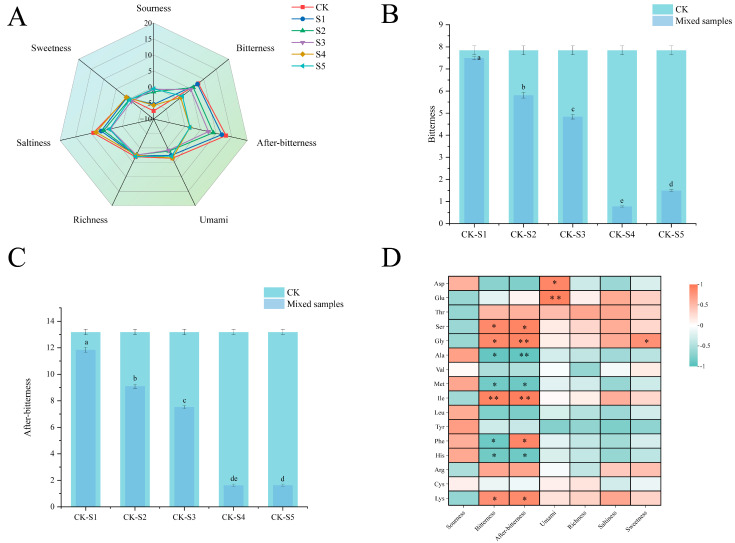
(**A**) E-tongue responses for FADS; (**B**) bitterness for CK and other composite seasonings; (**C**) after-bitterness values for CK and other composite seasonings; (**D**) Pearson correlation map of E-tongue and free amino acids. CK—DCB; S1—the BFA:DCB ratio was 3:7; S2—the BFA:DCB ratio was 4:6; S3—the BFA:DCB ratio was 5:5; S4—the BFA:DCB ratio was 6:4; S5—the BFA:DCB ratio was 7:3. Each square represents the Pearson correlation coefficient (R), where “*” indicates significant correlation (*p* < 0.05) and “**” indicates extremely significant correlation (*p* < 0.01); positive (0 < r < 1) and negative (−1 < r < 0) correlations are represented in orange and green, respectively. The different letters indicate significant differences (*p* < 0.05).

**Figure 5 foods-14-02976-f005:**
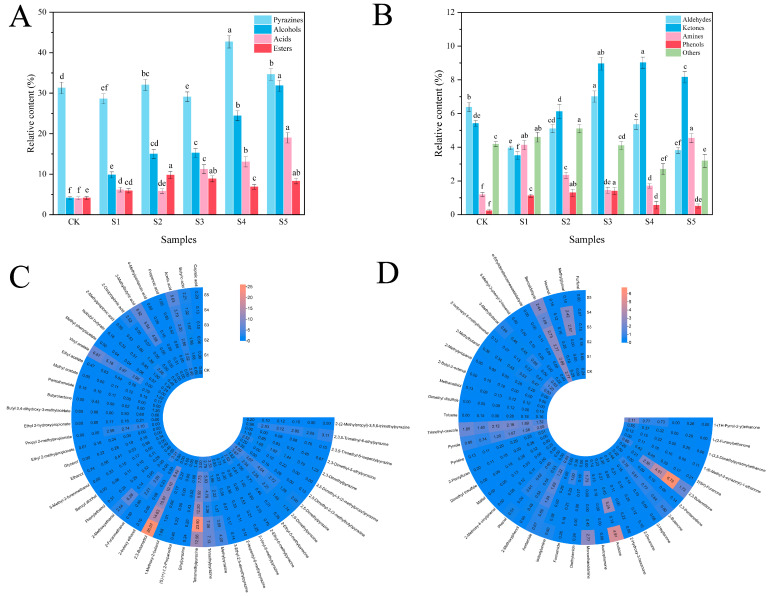
(**A**,**B**) Types and relative contents of volatile substances in FADS; (**C**,**D**) relative content of volatile flavor components in FADS. CK—DCB; S1—the BFA:DCB ratio was 3:7; S2—the BFA:DCB ratio was 4:6; S3—the BFA: DCB ratio was 5:5; S4—the BFA:DCB ratio was 6:4; S5—the BFA:DCB ratio was 7:3. The different letters indicate significant differences (*p* < 0.05).

**Figure 6 foods-14-02976-f006:**
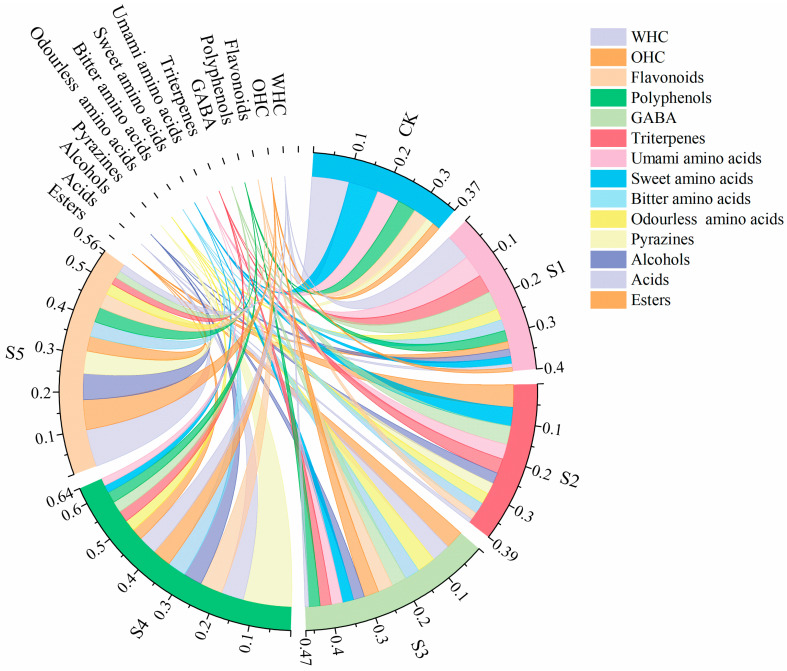
Graphs of quality scores for FADS. CK—DCB; S1—the BFA:DCB ratio was 3:7; S2—the BFA:DCB ratio was 4:6; S3—the BFA:DCB ratio was 5:5; S4—the BFA:DCB ratio was 6:4; S5—the BFA:DCB ratio was 7:3.

**Table 1 foods-14-02976-t001:** Free amino acid content of FADS (g/100 g DW).

Sample	CK	S1	S2	S3	S4	S5
Umami	0.46 ± 0.09 ^a^	0.46 ± 0.06 ^a^	0.43 ± 0.15 ^ab^	0.42 ± 0.11 ^ab^	0.40 ± 0.09 ^ab^	0.36 ± 0.04 ^ab^
Asp	0.03 ± 0.03 ^d^	0.10 ± 0.07 ^bc^	0.08 ± 0.01 ^bc^	0.10 ± 0.02 ^bc^	0.11 ± 0.01 ^bc^	0.12 ± 0.03 ^ab^
Glu	0.43 ± 0.08 ^a^	0.36 ± 0.11 ^ab^	0.35 ± 0.03 ^ab^	0.32 ± 0.01 ^c^	0.29 ± 0.21 ^cd^	0.24 ± 0.07 ^cd^
Sweet	0.65 ± 0.10 ^a^	0.57 ± 0.05 ^ab^	0.60 ± 0.11 ^ab^	0.58 ± 0.08 ^ab^	0.56 ± 0.06 ^ab^	0.53 ± 0.20 ^ab^
Thr	0.02 ± 0.01 ^a^	0.01 ± 0.01 ^ab^	0.01 ± 0.01 ^ab^	0.01 ± 0.01 ^ab^	0.01 ± 0.01 ^ab^	0.01 ± 0.00 ^ab^
Ser	0.17 ± 0.02 ^a^	0.11 ± 0.01 ^b^	0.11 ± 0.01 ^b^	0.09 ± 0.01 ^bc^	0.07 ± 0.00 ^d^	0.06 ± 0.02 ^de^
Gly	0.12 ± 0.05 ^a^	0.09 ± 0.06 ^ab^	0.09 ± 0.04 ^ab^	0.08 ± 0.02 ^ab^	0.07 ± 0.03 ^ab^	0.06 ± 0.03 ^ab^
Ala	0.34 ± 0.09 ^bc^	0.36 ± 0.05 ^bc^	0.39 ± 0.05 ^ab^	0.40 ± 0.15 ^ab^	0.41 ± 0.04 ^ab^	0.41 ± 0.15 ^ab^
Bitter	1.31 ± 0.23 ^bc^	1.66 ± 0.24 ^bc^	1.63 ± 0.16 ^bc^	1.72 ± 0.41 ^ab^	1.78 ± 0.22 ^ab^	1.75 ± 0.31 ^ab^
Val	0.23 ± 0.02 ^ab^	0.25 ± 0.03 ^ab^	0.24 ± 0.03 ^ab^	0.25 ± 0.14 ^ab^	0.26 ± 0.02 ^ab^	0.24 ± 0.03 ^ab^
Met	0.01 ± 0.01 ^e^	0.05 ± 0.02 ^cd^	0.05 ± 0.01 ^cd^	0.06 ± 0.03 ^bc^	0.07 ± 0.01 ^bc^	0.08 ± 0.01 ^b^
Ile	0.19 ± 0.03 ^a^	0.17 ± 0.07 ^ab^	0.17 ± 0.09 ^ab^	0.16 ± 0.07 ^ab^	0.15 ± 0.07 ^ab^	0.14 ± 0.02 ^ab^
Leu	0.34 ± 0.08 ^d^	0.59 ± 0.11 ^bc^	0.58 ± 0.04 ^bc^	0.63 ± 0.09 ^ab^	0.68 ± 0.09 ^ab^	0.68 ± 0.02 ^ab^
Tyr	0.27 ± 0.01 ^bc^	0.28 ± 0.02 ^bc^	0.28 ± 0.08 ^bc^	0.29 ± 0.08 ^ab^	0.28 ± 0.03 ^bc^	0.28 ± 0.12 ^bc^
Phe	0.21 ± 0.09 ^bc^	0.24 ± 0.02 ^bc^	0.24 ± 0.02 ^bc^	0.25 ± 0.02 ^bc^	0.26 ± 0.11 ^ab^	0.26 ± 0.11 ^ab^
His	0.02 ± 0.01 ^ab^	0.03 ± 0.01 ^ab^	0.03 ± 0.02 ^ab^	0.04 ± 0.05 ^a^	0.04 ± 0.01 ^a^	0.04 ± 0.01 ^a^
Arg	0.04 ± 0.03 ^ab^	0.05 ± 0.02 ^a^	0.04 ± 0.02 ^ab^	0.04 ± 0.01 ^ab^	0.04 ± 0.02 ^ab^	0.03 ± 0.01 ^ab^
Odourless	0.25 ± 0.12 ^a^	0.16 ± 0.08 ^ab^	0.16 ± 0.11 ^ab^	0.14 ± 0.12 ^ab^	0.12 ± 0.04 ^ab^	0.10 ± 0.05 ^ab^
Cys	0.01 ± 0.00	0.01 ± 0.01	0.01 ± 0.01	0.01 ± 0.01	0.01 ± 0.00	0.01 ± 0.00
Lys	0.24 ± 0.02 ^a^	0.15 ± 0.04 ^b^	0.15 ± 0.06 ^b^	0.13 ± 0.07 ^bc^	0.11 ± 0.12 ^bc^	0.09 ± 0.04 ^bc^
Essential Amino Acids	1.24 ± 0.03 ^c^	1.46 ± 0.15 ^ab^	1.45 ± 0.15 ^ab^	1.49 ± 0.13 ^ab^	1.54 ± 0.23 ^ab^	1.50 ± 0.20 ^ab^
Total Amino Acids	2.67 ± 0.31 ^ab^	2.85 ± 0.35 ^ab^	2.82 ± 0.46 ^ab^	2.86 ± 0.38 ^ab^	2.86 ± 0.32 ^ab^	2.74 ± 0.37 ^ab^

Note: CK—DCB; S1—the BFA:DCB ratio was 3:7; S2—the BFA:DCB ratio was 4:6; S3—the BFA:DCB ratio was 5:5; S4—the BFA:DCB ratio was 6:4; S5—the BFA:DCB ratio was 7:3. The different letters indicate significant differences (*p* < 0.05).

## Data Availability

Data are contained within the article.

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
