# Peer review of "Synergistic Flavor Modulation and Functional Enhancement of Douchiba via Compounding with *Bacillus subtilis*-Fermented Adlay"

_foods, 2025, doi:10.3390/foods14172976_

Round 1

Reviewer 1 Report

Comments and Suggestions for Authors Before consider the puclication of your MS, you should make some changes:
Make sure all figures are properly referenced in the text.
Include in the graphs the information on the statistical analyzes performed to show significant differences.
The language is acceptable but needs to be edited for conciseness.
You may consider including your fermentation data to determine how the different fermentations promoted the increase or decrease of the different ingredients. It is very valuable to include this data.

Line: 3 and 18: Write “Bacillus” insted B., write the full name the first time that it appead.

Line 65: “Bacillus subtilis”. Please carefully review the proper use of the italiks along the text.

Line 21: Replace “via the Entropy Method” with “via the Entropy Method for composite scoring”.

Line 62: Rephrase to improve clarity. Change “requiring resolution” for “that needs to be addressed”.

Line 84: You can consider to improve the connection between the aim and significance.

Line 103: Clarify storage condition purpose. For example: “Samples were stored at −4 °C to preserve volatile and functional compounds prior to analysis.”

Line 105: If you have the data of the controls, you can use them to determine the ΔE. You can consider use the data before the fermentation. Ideally you should have that data.

Line 126: Ensure consistency in formula notation; use either capital or lowercase letters uniformly. Use the equation format for this formula.

Line 161: Include the information regading the post hod test used.

Line 169: Figures are referenced as “Error! Reference source not found.” Reviwe this along all the text.

Line 179-180: Rewrite the sentence to improve the clarity.

Line 202–203: Sentence is convoluted, rewrite it to improve the clarity.

Line 246–247: You can referencing comparative bioavailability or relevance to health claims.

Line 268: Add citation after mentioning amino acid complementation.

Line 294: Strengthen the explanation of this sentence.

Lines 337–404: This section is highly informative but dense. Consider clearer segmentation.

Line 342: Rewrite the sentence to make it clear.

Line 425: Review the numeration.

Line 427: Modify “achieving dual optimization” with “successfully improving both sensory attributes and functional properties.”

Line 439–440: Rephrase for clarity. Comments on the Quality of English Language

I'm not an EN native speaker, but some sentences are too long and complex to read; therefore, I consider that performing the style revision can improve the quality of the paper.

Author Response

Comments 1: Make sure all figures are properly referenced in the text.
Response 1: Thank you for pointing this out. We have re-cited the charts in the article correctly.
Comments 2: Include in the graphs the information on the statistical analyzes performed to show significant differences.
Response 2: Thank you for pointing this out. We agree with this comment. Therefore, we have modified all images in the article that were not marked as significant differences (Figure 2B, page 5; Figure 4B and Figure 4C, page 12). 
Comments 3: The language is acceptable but needs to be edited for conciseness.
Response 3: I would like to thank the reviewer for your comments. We have simplified the more complex sentences in the article.
Comments 4: You may consider including your fermentation data to determine how the different fermentations promoted the increase or decrease of the different ingredients. It is very valuable to include this data.
Response 4: Thank you for your valuable comments on this paper. Your suggestion to supplement fermentation data to analyze the effects of different fermentation conditions on component changes is highly valuable. Unfortunately, due to the technology limitations of this study, we were unable to systematically monitor the dynamic changes in component data during the fermentation process. Nevertheless, the primary focus of this study was on the functional properties and flavor characteristics of the final fermented composite product, and the existing data are sufficient to support this conclusion. In future research, we will prioritize the design of a more comprehensive fermentation process monitoring scheme to further elucidate the kinetic mechanisms underlying component changes.
Comments 5: Line: 3 and 18: Write "Bacillus" instead of B., write the full name the first time that it appead.

Response 5: We sincerely thank the reviewers for their valuable comments. As suggested, we have revised the manuscript to ensure that the full genus name "Bacillus" is used at its first mention in the text (page 1).
Comments 6: Line 65: "Bacillus subtilis". Please carefully review the proper use of the italiks along the text.
Response 6: We sincerely appreciate your valuable feedback. Based on your suggestion, we have made the corresponding changes in the text (page 2).
Comments 7: Replace "via the Entropy Method" with "via the Entropy Method for composite scoring".
Response 7: Thank you for your valuable suggestion regarding the precision of the manuscript's wording. As recommended, we have revised the original phrase "via the Entropy Method" to "via the Entropy Method for composite scoring" (page 1).
Comments 8: Line 62: Rephrase to improve clarity. Change "requiring resolution" for "that needs to be addressed".
Response 8: We appreciate the valuable comments provided by the reviewers. Based on your suggestion, we have revised the original phrase "requiring resolution" to "that needs to be addressed" (page 2).

Comments 9: Line 84: You can consider to improve the connection between the aim and significance.
Response 9: Thank you very much for your careful review of the manuscript and your valuable comments. Regarding the issue you pointed out in line 84, "consider improving the connection between the purpose and significance," we have conducted in-depth consideration and analysis. Based on this, we have reorganized and revised the relevant content as follows: "Finally, a quality evaluation system was established using the Entropy Method to determine the optimal BFA to DCB ratio that achieved the best balance between physicochemical properties and sensory acceptability. The study not only resolved the contradiction between functional enhancement and flavor degradation, but also provided a scientific basis for the development of a new generation of fermented condiments.". Please refer to the text for details (page 2).
Comments 10: Line 103: Clarify storage condition purpose. For example: "Samples were stored at −4 ℃ to preserve volatile and functional compounds prior to analysis."
Response 10: Thank you very much for pointing out the need to clarify the purpose of the storage conditions in line 103 of the manuscript. I sincerely apologize for an error in the original submission where the temperature was incorrectly stated as "-4℃" instead of the correct value "4℃". 
We have now modified the following sentence in the text: "And the obtained compound seasonings were stored for later use at 4°C in the refrigerator to preserve volatile and functional compounds prior to analysis. " (page 3).

Comments 11: If you have the data of the controls, you can use them to determine the ΔE. You can consider use the data before the fermentation. Ideally you should have that data.
Response 11: Dear Reviewer, we sincerely thank you for your attention and valuable suggestions on the method for determining the ΔE value in this study. Your idea of using control group data and pre - fermentation data to determine it is scientific and reasonable, offering vital guidance for improving result accuracy and reliability.
However, our study mainly centers on the final sensory traits of the product. Li et al. (2021), in their 2021 article on calculating total color difference in fermented products, also didn't use unfermented data for calculations.
So, we think it's also acceptable to not use unfermented data for ΔE value determination for now. Thanks again for your suggestions.
Li, X.; Gao, J.; Jesus, S.; Wang, X.; Giovanni, C.; Si, M.; Sang, Y. Effect of fermentation by Lactobacillus acidophilus CH-2 on the enzymatic browning of pear juice. Lwt. 2021, 147. http://dx.doi.org/10.1016/J.LWT.2021.111489
Comments 12: Ensure consistency in formula notation; use either capital or lowercase letters uniformly. Use the equation format for this formula.
Response 12: Thank you very much for pointing out the need to ensure consistency and uniform use of upper and lower case letters in the formulas in the manuscript. We immediately checked all the formulas and found that there were indeed cases of mixed use of symbols. We have now standardized the formulas by changing all letters to upper case. We have used the equation format for this formula (page 3).

Comments 13: Line 161: Include the information regading the post hod test used.
Response 13: We sincerely thank the reviewer for your careful review of this paper and your valuable comments. We have supplemented the relevant content and added detailed information on the hod test used: SPSS 27.0 software (SPSS Inc., Chicago, IL, USA); Origin 2024 software (OriginLab Corporation, Northampton, MA, USA) (page 4).
Comments 14: Figures are referenced as “Error! Reference source not found.” Review this along all the text.
Response 14: We sincerely apologize for the issue where the cited content in the figure displayed "Error! Reference source not found" in the manuscript. This was our oversight, and we deeply apologize for any inconvenience caused. Upon receiving the feedback, we immediately conducted a thorough review of the cited content and made the necessary revisions.
Comments 15: Line 179-180: Rewrite the sentence to improve the clarity.
Response 15: We sincerely appreciate your careful review of lines 179–180 of our manuscript and your reminder that we need to rewrite them to improve clarity. We recognized that the original sentences were problematic. We have revised the original sentence according to your suggestion as follows: " Especially in mixed samples with a high content of BFA (S3, S4, and S5), the difference was more pronounced. Overall, this change improved the sensory acceptability of the compound seasoning to some extent."(page 5).

Comments 16: Line 202–203: Sentence is convoluted, rewrite it to improve the clarity.
Response 16: We sincerely appreciate your careful review of our manuscript. We apologize for any difficulty readers may have encountered due to the complex sentence structure. We recognize that the original wording was unclear. We have revised the original sentence according to your suggestion as follows: " A small amount of DCB could improve lipid binding by increasing hydrophobicity. However, when the proportion of DCB was too high, it caused excessive protein aggregation, resulting in the formation of dense structures that hindered lipid penetration." (page 5).
Comments 16: Line 202–203: Sentence is convoluted, rewrite it to improve the clarity.
Response 16: We sincerely appreciate your careful review of our manuscript. We apologize for any difficulty readers may have encountered due to the complex sentence structure. We recognize that the original wording was unclear. We have revised the original sentence according to your suggestion as follows: " A small amount of DCB could improve lipid binding by increasing hydrophobicity. However, when the proportion of DCB was too high, it caused excessive protein aggregation, resulting in the formation of dense structures that hindered lipid penetration." (page 5).

Comments 17: Line 246–247: You can referencing comparative bioavailability or relevance to health claims.
Response 17: We sincerely appreciate your careful review of our manuscript. We have added relevant content to the corresponding parts in the text: " GABA could reduce pain and anxiety; therefore, it could treat neurological disorders such as depression. In addition, it also activates GABAA and GABAB receptors in pancreatic β cells. Thereby, it could promote insulin release and treat diabetes (Zhang et al., 2022). Triterpenoids, particularly lupinol and betulin, exhibit strong anti-inflammatory and antibacterial properties (Golubova et al., 2025). Therefore, FADS, which is rich in triterpenoids, could compensate for its low flavonoids and polyphenols content. Overall, adding BFA enhanced the physiological activities of the seasoning, such as blood pressure reduction and anti-anxiety effects. And strengthening its anti-inflammatory and anti-cancer properties." (page 6-7).
Zhang, Y.;     Zhang, M.; Li, T.; Zhang, X.; Li, W. Enhance Production of γ-Aminobutyric Acid (GABA) and Improve the Function of Fermented Quinoa by Cold Stress. Foods. 2022, 11, 3908. http://dx.doi.org/10.3390/FOODS11233908
Golubova, D.; Salmon, M.; Su, H.; Tansley, C.; Kaithakottil, G.G.; Linsmith, G.; Schudoma, C.; Swarbreck, D.; Connell, M.O.; Patron, N.J. Biosynthesis and bioactivity of anti-inflammatory triterpenoids in Calendula officinalis. Nat. Commun. 2025, 16, 6941. http://dx.doi.org/10.1038/S41467-025-62269-W

Comments 18: Line 268: Add citation after mentioning amino acid complementation.
Response 18: We sincerely appreciate your valuable feedback. Based on your suggestion, we have added a citation after mentioning amino acid complementation: "The amino acids in grains and legumes are usually complementary (Han et al., 2021). Thus, the content of essential amino acids in FADS was more balanced as a result of amino acid complementation between BFA (methionine and leucine) and DCB (lysine)." (page 7).
Han, F.; James, M.P.; Juntao, L.; Natascha, S.; Pang, S. The Complementarity of Amino Acids in Cooked Pulse/Cereal Blends and Effects on DIAAS. Plants. 2021, 10, 1999. http://dx.doi.org/10.3390/PLANTS10101999

Comments 19: Line 294: Strengthen the explanation of this sentence.
Response 19: Thank you very much for pointing out that the sentence in line 294 needs further explanation. We recognize that the original wording was not comprehensive enough. We have explained the sentence based on your comments as follows: "Hydrophobic peptides, bitter amino acids, isoflavonoids, and saponins are common compounds with bitter properties. The bitterness of FADS was directly correlated with DCB proportion because these substances were produced and increased in the fermentation of DCB.” (page 10). 
Comments 20: Lines 337–404: This section is highly informative but dense. Consider clearer segmentation.
Response 20: I would like to thank the reviewer for your comments. Based on your suggestion, we have divided this section into clearer paragraphs (page 12-13).

Comments 21: Line 342: Rewrite the sentence to make it clear.
Response 21: Thank you very much for pointing out that the manuscript needs to be rewritten to improve clarity. We have recognized that the original sentence was unclear. We made revisions after receiving your feedback as follows: "Among the five composite seasoning sample groups, pyrazines exhibited the highest relative content (28.60%–42.68%), followed by alcohols (4.11%–31.84%) and acids (4.04%–18.98%). These findings indicate that pyrazines, alcohols, and acids constituted the primary flavor-contributing compounds in FADS."(page 11).
Comments 22: Line 425: Review the numeration.
Response 22: Thank you for pointing out my mistake. I am very sorry for my error. We have already made the correction in the original text as follows: 4. conclusion (page 14).
Comments 23: Line 427: Modify “achieving dual optimization” with “successfully improving both sensory attributes and functional properties.”
Response 23: We appreciate the valuable comments provided by the reviewers. Based on your suggestion, we have revised the original phrase "achieving dual optimization" to "successfully improving both sensory attributes and functional properties" (page 14).

Comments 24: Line 439–440: Rephrase for clarity.
Response 24: Thank you for your suggestion. This was an oversight on our part, and we apologize for it. We have revised the relevant content as follows: "In conclusion, our research showed that FADS improved both functional and sensory qualities in concert. This improvement stemmed from complementary amino acids, bidirectional taste compensation, and synergistic effects of flavor components between BFA and DCB. These findings provide a fundamental theoretical basis for optimizing flavor profiles in fermented seasonings."(page 14).
3. Response to Comments on the Quality of English Language
Point 1: The language is acceptable but needs to be edited for conciseness.
Response 1: Thank you for your helpful suggestion. We have carefully revised the manuscript to improve the clarity and fluency of the English. The changes include corrections in grammar, sentence structure, and overall readability. We hope this revised version meets the required language standards.

Reviewer 2 Report

Comments and Suggestions for Authors

Dear authors, first of all, I'd like to congratulate you on your work. However, I believe that in the attached file, I propose improvements that should be applied to the manuscript to further highlight its existing quality. Thank you very much.

Author Response

Comments 1: Keywords should not be the same as the title, this makes that you are losing potential when researchers search for you. Therefore, "Douchiba" and "fermented adlay" should be modified with other keywords, such as, as suggested, "Entropy Method" or "E-tongue".

Response 1: Thank you for your valuable comments on this paper. Based on your suggestion, we have changed “Douchiba” to “Entropy Method and "fermented adlay" to "E-tongue". The corresponding modifications have been made to the Keywords section in the original text. (page 1).

Comments 2: Line 50-51: There is no reference to support these beneficial properties, please add.

Response 2: We sincerely appreciate your valuable feedback. Based on your suggestion, we have added citation to support these beneficial properties (page 2).

  Huang, Y.; Li, T.; Han, X.; Qiu, X. Processing of Chili Beef Paste with Flavor Douchi Cake (China). Farm Prod. Proc. 2021, 1-4. http://dx.doi.org/10.16693/j.cnki.1671-9646(X).2021.11.034.

Comments 3: Line 102: Was the powder obtained from the different combinations stored in a vacuum bag? And why wasn't it placed in a freezer at -18°C or -80°C to completely eliminate any trace of moisture?

Response 3: I would like to thank the reviewer for your comments. I sincerely apologize for an error in the original submission where the temperature was incorrectly stated as "-4℃" instead of the correct value "4℃".

Here are my answers to your questions: 1. The powder obtained from the different combinations were stored in vacuum bags. This was not indicated in the article, which was my mistake. We have added “The powder obtained from the different combinations was stored in vacuum bags.” to the article (page 3). 2. The majority of tests are short-term. Thus, the current storage method (4°C) satisfies quality standards.

Comments 4: Section 2.9.: It is missing to add which column is used. It remains to be added which column is used, and of course, what brand and model of GC-MS it is. I think the description of materials and methods in the volatiles section is too brief for all the components of that analysis. Please expand on this information.

Response 4: Thank you very much for your attention to Section 2.9 of the manuscript and for pointing out the issue. The name of the chromatographic column is: DB-Wax Chromatography Column. The brand and model of GC-MS are: Pegasus BT, LECO, USA. We have supplemented the relevant information and description of the detection method in the text (page 4), as follows:

Volatile compounds were determined according to the previous study with some slight modifications (Wang et al., 2023).

  Volatile compounds in FADS were analyzed based on a HS-SPME-GC-MS (Pegasus BT, LECO, USA). With an aged 50/30 μm CAR/PDMS/DVB extraction head, the samples were separated into 20 mL headspace vials, adsorbed for 30 min at 60°C, and then desorbed for 3 min at 250°C. Finally, the data were acquired.

  GC conditions were set as follows: DB-Wax Chromatography Column (30 m × 0.25 mm, 0.25 μm). Heating program: initial temperature (40°C) for 3 min, final temperature in the oven (230°C) for 5 min, and heating rate (10 °C/min).

  MS conditions were set as follows: ionization mode (El+), electron energy (70 eV), surface temperature (200°C), interface temperature (250°C), detector voltage (2000 V), and emission current (1 mA).

Wang, Q.; Wen A., Qin, L., Hu, Y.; Zhu Y. Study on Characteristic Flavor Substances in Traditional Fermented Douchiba. Food Ferment. Sci. Technol (China). 2023, 59, 62-72. http://dx.doi.org/10.3969/j.issn.1674-506X.2023.01-009

Comments 5: Section 2.10.: The letters in the graphs express the statistically significant

difference between samples, however, they have not included in this section what statistical method they used, Tukey, Duncan, etc. Nor has it been added what statistically significant level was used to determine the differences between samples, e.g.: p<0.5 or p<0.1, etc. Please add the required information.

Response 5: Thank you very much for your careful review of Section 2.10. You have accurately pointed out the lack of statistical information in this section, which is crucial for improving the scientific and rigorous nature of the research. We have added the following information: “The differences among the results were analyzed by Duncan’s test and analysis of variance (ANOVA) (P<0.05).” (page 4).

Comments 6: Throughout the results section, there's a phrase that says: "Error! Reference source not found. A," B, or C; or just the phrase itself, without the capital letters. Obviously, this isn't part of the text; please remove all of these errors. No paragraph has tables or figures associated with it to display the results. I believe this is because the message that appears multiple times replaces the reference made by the authors to the tables or figures.

Response 6: We sincerely apologize for the issue where the cited content in the figure displayed “Error! Reference source not found” in the manuscript. This was our oversight, and we deeply apologize for any inconvenience caused. Upon receiving the feedback, we immediately conducted a thorough review of the cited content and made the necessary revisions.

Comments 7: Section 3.4: In my opinion, this section is poorly expressed, confusing the reader with the distinction between essential and free amino acids, which can be both at the same time, as is the case with leucine. Please clarify this point.

Furthermore, 6 of the 8 amino acids associated with bitterness increased in the different samples. Regarding sweetness, only 1 of the 4 associated amino acids increased, while the other 3 decreased.

Therefore, NO, the presence of bitter amino acids did NOT improve umami intensity. The table clearly shows how, as bitterness intensity increases in the different samples, unami decreases; it is inversely proportional.

Please rewrite the paragraph, taking into account the data shown.

Response 7: Thank you very much for your careful review of Section 3.4. I apologize for any confusion caused by the description in this section. Firstly, free amino acids include essential amino acids. Moreover, all essential amino acids can exist in a free form. Therefore, leucine can be referred to as both an essential amino acid and a free amino acid. We have enhanced the explanation of free amino acids in the text as follows: “Free amino acids include both essential amino acids and non-essential amino acids”(page 7).         Secondly, it is true that bitter amino acids do not enhance the intensity of umami flavor; they merely add richness to the taste perception. This was our oversight, and we sincerely apologize for it. The corresponding section in the text has also been revised accordingly as follows: “Although the content of bitter amino acids is relatively high, the presence of bitter amino acids plays an important role in prolonging taste, and enhancing complexity and richness” (page 7).

Comments 8: Line 284: Please start the section properly.

Response 8: Thank you for pointing out my mistake. I am very sorry for my error. We have already made the correction in the original text (page 10).

Comments 9: Figure 5 C: First, the figure is very small even when enlarged, making the data impossible to read.Also, why are only a portion of the volatile compounds listed, and not all 89? I haven't read any criteria for this.

Therefore, I recommend changing the figure for better understanding and completing it (unless there's some reason) with the missing volatiles.

Response 9: We sincerely apologize for the poor quality of the figures in the initial submission. This is entirely our responsibility and reflects an oversight during the final compilation and file export process. We understand that clear and legible figures are essential for evaluating the results and we deeply regret that the presented figures did not meet this standard, hindering your review. We have re-uploaded higher resolution images to ensure that our data is accurately and clearly presented.

I sincerely appreciate your thoughtful analysis and insightful recommendations about how to present volatile chemicals in this study. The issue you pointed out, namely that only a portion of volatile compounds were listed without specifying the screening criteria, is indeed a shortcoming in our preliminary work. The reason why not all compounds were listed in the first manuscript was that the content of these substances was relatively low, so I included them in the appendix. Based on your suggestion, I have presented them in the form of a figure (Figure 5D; page 13 ).

Comments 10: The word "Conclusions" is offset and not bold. Please normalize the text.

Response 10: Thank you for pointing out my mistake. We have already made it bold and unified the format in the article (page 14).

Round 2

Reviewer 2 Report

Comments and Suggestions for Authors

Dear authors,

Thank you for including the suggested changes. However, on lines 177 and 179, the Error! Reference source not found. A and B, respectively, comment appears again.

Please correct it.

Congratulations for this manuscript.

Thank you very much.

Author Response

Comments 1: On lines 177 and 179, the Error! Reference source not found. A and B, respectively, comment appears again.
Response 1: Thank you for your valuable comments on this paper. Based on your suggestion, We have re-cited the charts in the article correctly (page 1).
